# Generative Factor Chaining: Coordinated Manipulation with Diffusion-based Factor Graph

**Utkarsh A. Mishra, Yongxin Chen, Danfei Xu**
Georgia Institute of Technology
{umishra31, yongchen, danfei}@gatech.edu

**Abstract:** Learning to plan for multi-step, multi-manipulator tasks is notoriously difficult because of the large search space and the complex constraint satisfaction problems. We present Generative Factor Chaining (GFC), a composable generative model for planning. GFC represents a planning problem as a spatial-temporal factor graph, where nodes represent objects and robots in the scene, spatial factors capture the distributions of valid relationships among nodes, and temporal factors represent the distributions of skill transitions. Each factor is implemented as a modular diffusion model, which are composed during inference to generate feasible long-horizon plans through bi-directional message passing. We show that GFC can solve complex bimanual manipulation tasks and exhibits strong generalization to unseen planning tasks with novel combinations of objects and constraints. More details can be found at: generative-fc.github.com

**Keywords:** Manipulation Planning, Bimanual Manipulation, Generative Models

## 1 Introduction

Solving real-world sequential manipulation tasks requires reasoning about sequential dependencies among manipulation steps. For example, a robot needs to grip the center or the tail of a hammer, instead of its head, in order to subsequently hammer a nail. The complexity of planning problems increases when multiple manipulators are involved, where spatial coordination constraints among manipulators need to be satisfied. In the example shown in Figure 1, the robot has to reason about the optimal pose to grasp the hammer with the left arm, such that the right arm can coordinate to re-grasp. Subsequently, the two arms must coordinate to hammer the nail. While classical Task and Motion Planning (TAMP) methods have shown to be effective at solving such problems by hierarchical decomposition [1], they require accurate system state and kinodynamic model. Further, searching in such a large solution space to satisfy numerous constraints poses a severe scalability challenge. In this work, we aim to develop a learning-based planning framework to tackle complex manipulation tasks with both sequential and spatial coordination constraints.

To solve complex sequential manipulation problems, prior learning-to-plan methods have largely adopted the options framework and modeled the preconditions and effect of the options or primitive skills [2, 3, 4, 5, 6, 7]. Key to their successes are skill chaining functions that determine whether executing a skill can satisfy the precondition of the next skill in the plan, and eventually the success condition of the overall task. However, the use of vectorized states and the assumption of a linear chain of sequential dependencies limits the expressiveness of these methods. Consider a task where a robot fetches two items from a box. Intuitively, the skills for fetching one object should not influence the other. However, due to vectorized states and the linear dependency assumption, the skill-chaining methods are forced to model such sequential dependencies. Similarly, a skill intended to satisfy a future skill's condition will be forced to influence the steps in between. Finally, the skill chain representation forbids these methods from effectively modeling multiple-arm manipulation tasks, where concurrent skills must be planned to jointly satisfy a constraint.

8th Conference on Robot Learning (CoRL 2024), Munich, Germany.

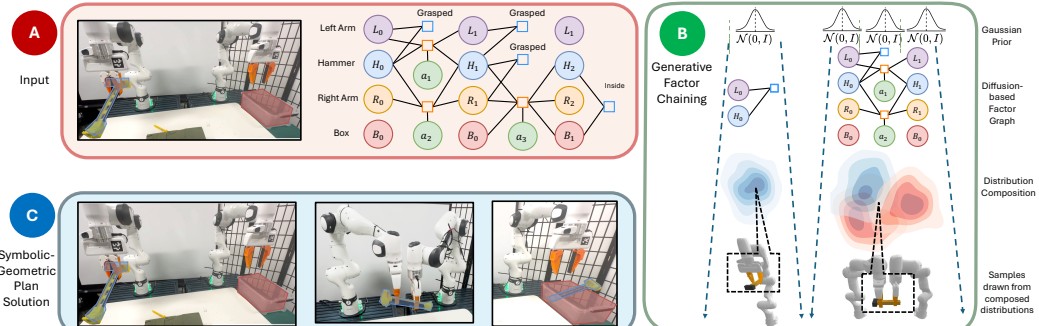

Figure 1: **Factor graph for a multi-arm coordination task.** Our factor graph-based planning formulation solves for a sequence of spatial factor graphs from the initial state to a goal factor by chaining them using temporal skill factors. **Task:** The task objective is to place the hammer inside the box. However, since the left arm cannot reach the box, the hammer is handed over to the right arm such that the right arm can complete the task. **(a) Inputs:** The initial scene and a symbolically feasible spatial-temporal factor graph plan to complete the goal objective. **(b) GFC:** We formulate all factors as distributions of the nodes connected to them. GFC represents spatial factors as classifier-distributions and temporal factors as diffusion model learned distributions. We leverage compositionality of diffusion models to compose spatial-temporal distributions and find the joint distribution of the complete plan directly at inference. **(c) Output:** Finally, samples drawn from such a joint distribution are symbolically and geometrically feasible solutions of the whole plan.

To move beyond the linear chain and model complex coordinated manipulation, we introduce Generative Factor Chaining (GFC), a learning-to-plan framework built on flexible composable generative models. For a given symbolically feasible plan graph, GFC adopts a spatial-temporal factor graph [8] representation, where nodes are objects and robot states, and spatial factors represent the relationship constraints between these nodes. Skills are temporal factors that connect these state-factor graphs via transition distributions. A single skill factor can simultaneously connect to multiple object and robot nodes, allowing for natural representation of complex multi-object interactions and steps that necessitate coordination between multiple manipulators. During inference, this factor graph can be treated as a probabilistic graphical model, where the learned skill factor and spatial constraint factor distributions are composed to form a joint distribution of complete plans. Through 13 long-horizon manipulation tasks in simulation and the real world, we show that GFC can solve complex bimanual manipulation tasks and exhibits strong generalization to unseen planning tasks with novel combinations of objects and constraints.

## 2 Related Work

**Task and Motion Planning (TAMP).** TAMP frameworks decompose a complex planning problem into constraint satisfaction problems at task and motion levels [9, 2, 10, 11, 12]. Notably, Garret et al. [1] drew connections between TAMP and factor graphs [8], representing constraints as factors and objects/robots as nodes. This formalism naturally allows reusing per-constraint solvers across tasks. While classical TAMP relies on accurate perception and system dynamics, limiting scalability, we take a learning approach, though our compositional factor graph representation remains inspired by classical TAMP.

**Generative models for planning.** Modern generative models have been applied to offline imitation [13, 14, 15, 16, 17, 18, 19, 20] and reinforcement learning [21, 22]. In addition to modeling complex state and action distributions, generative models have also been shown to encourage compositional generalization [23, 6, 24] by combining data across tasks [22, 21]. Most relevant to us are Generative Skill Chaining (GSC) [6] and Diffusion-CCSP [25], both designed to achieve systematic compositional generalization. GSC composes skill chains through a guided diffusion process but fail to solve non-sequential dependencies similar to other skill-chaining methods [4, 5]. Diffusion-CCSP trains diffusion models to generate configurations to satisfy spatial constraints and use exter-

nal solvers to plan the manipulation sequence. Our method is a unified framework and generates motion plans to satisfy both spatial and temporal constraints represented in a factor graph.

**Learning for coordinated manipulation.** Coordinating two or more arms for manipulation presents numerous planning challenges [26, 27, 28], including the combinatorial search space complex constraints for coordinated motion. Recent works have utilized learning-based frameworks [29, 30, 31, 32, 33] in both Reinforcement Learning [29, 31] and offline Imitation Learning [33, 32]. However, most existing works have focused on learning task-specific policies [29, 32] or require multi-arm demonstration data collected through a specialized teleoperation device [33]. In contrast, our factor graph-based representation enables solving multi-arm tasks by composing multiple single-arm skills through inference-time optimization.

## 3 Background

**Diffusion Models.** A core component of our method is based on distributions learned using diffusion models. A diffusion model learns an unknown distribution $p(\mathbf{x}^{(0)})$ from its samples by approximating the score function $\nabla \log p$. It consists of two processes: a *forward diffusion or noising* process that progressively injects noise and a *reverse diffusion or denoising* process that iteratively removes noise to recover clean data. The forward process simply adds Gaussian noise $\epsilon$ to clean data as $\mathbf{x}^{(t)} = \mathbf{x}^{(0)} + \sigma_t \epsilon$ for a monotonically increasing $\sigma_t$. The reverse process relies on the score function $\nabla_{\mathbf{x}} \log p_t(\mathbf{x}^{(t)})$ where $p_t$ is the distribution of noised data $\mathbf{x}^{(t)}$. In practice, the unknown score function is estimated using a neural network $\epsilon_\phi(\mathbf{x}^{(t)}, t)$ by minimizing the denoising score matching [34] objective $\mathbb{E}_{t,\epsilon,\mathbf{x}^{(0)}}[\lambda(t)\|\epsilon - \epsilon_\phi(\mathbf{x}^{(t)}, t)\|^2]$ where $\lambda(t)$ is a time-dependent weight. Several recent works have explored the advantages of diffusion models like scalability [35, 36, 37, 38] and the ability to learn multi-modal distributions [39, 40, 41, 22]. We are particularly interested in the compositional ability [23, 13, 24, 25, 6] of these models for the proposed method.

**Problem setup.** We assume access to a library of parameterized skills [42] $\pi \sim \Pi$ such as primitive actions like `Pick` and `Place`. Each skill $\pi$ requires a pre-condition to be fulfilled and is parameterized by a continuous parameter $a \in A_\pi$ governing the desired motion while executing the skill in a state $s$. For a given symbolically feasible task plan from a starting state $s_0$ to reach a specified goal condition $s_{goal}$, generated by a task planner or given by an oracle, the problem is to obtain the sequence of continuous parameters to make the plan geometrically feasible. For example, given a nail at a target location and a hammer on a table, the symbolic plan is to `Pick` the hammer and `Reach` the nail. A geometrically-feasible plan requires suitable `Pick` and `Reach` parameters such that the hammer's head can strike the nail.

**Learning for skill chaining.** Existing works along this direction model the planning problem as a "chaining" problem: They first model the pre-conditions and effect state distributions for every skill $\pi \sim \Pi$ from the available data and a symbolic *plan skeleton* $\Phi_K = \{\pi_1, \pi_2, ..., \pi_K\}$ consisting of $K$-skills is constructed. With this model, they search for the given skill sequence (plan) such that each skill satisfies the pre-conditions of the next skill in the plan. STAP [5] used learned priors to perform data-driven optimization with the cross-entropy maximization method. In GSC [6], the policy and transition model is formulated as a diffusion model based distribution $p_\pi(s, a_\pi, s')$ which allows for flexible chaining. While the forward chain ensures dynamics consistency in the plan, backward chain ensures that the goal is reachable from the intermediate states. For a forward rollout trajectory $\tau = \{s_0, a_{\pi_1}, s_1, a_{\pi_2}, s_{goal}\}$ associated with skeleton $\Phi_2 = \{\pi_1, \pi_2\}$, the resulting forward-backward combination based on GSC [6] can be represented as

$$p_\tau(\tau|s_0, s_{goal}) \propto \frac{p_{\pi_1}(s_0, a_{\pi_1}, s_1)p_{\pi_2}(s_1, a_{\pi_2}, s_{goal})}{\sqrt{p_{\pi_1}(s_1)p_{\pi_2}(s_1)}} \tag{1}$$

## 4 Method

We aim to solve unseen long-horizon planning problems by exploiting the inter-dependencies between the objects relevant for the task at hand. Our method adopts factor graphs to represent states and realize their temporal evolution by the application of skills. While previous works have considered *vectorized state* representations making it difficult to decouple spatial-independence, we focus

on *factorized state* representations such that the state of the environment is entirely modular, containing information about all the objects in the scenario and the task-specific constraints between them. We use a spatial-temporal factor graph [8] that is transformed into a probabilistic graphical model by representing temporal factors as skill-level transition distributions and spatial factors as constraint-satisfaction distributions. A composition of all the factors jointly represents sequential and coordinated manipulation plans directly at inference and can be solved by sampling optimal node variables using reverse diffusion sampling.

## 4.1 Representing States, Skills, and Plans in Factor Graphs

**States as factor graphs.** We define a factor graph $\{\mathcal{V}, \mathcal{F}\}$ of a state $s$ consisting of the decision variable $\mathcal{V}$ and factor $\mathcal{F}$ nodes. Every robot and object is represented as a decision variable node $v \in \mathcal{V}$ containing their respective state. Factors $f \in \mathcal{F}$ between nodes in a given state are *spatial constraints*. For example, a `Grasped` spatial factor specifies admissible rigid transforms between a gripper and an object. When we construct a probabilistic graphical model from the representation described above, an intuitive way of calculating the distribution of a state, $p(s)$, is the composition of all the factor distributions. Mathematically:

$$p(s) \propto \prod_{f \in \mathcal{F}} p_f(\mathcal{S}_f) \quad \text{where } s \equiv \bigcup_{f \in \mathcal{F}} \mathcal{S}_f \tag{2}$$

where $p_f(\mathcal{S}_f)$ represents the joint factor potential of nodes $v \in \mathcal{S}_f \subseteq \mathcal{V}$, i.e. all nodes involved in a factor [1] and $s$ is the joint distribution of all such nodes. This indicates that the joint distribution of all the nodes must satisfy each of the factors, also explored by Diffusion-CCSP [25].

**Skills as temporal factors.** To represent transitions between states, we adapt parameterized skills [42] for a factor graph formulation. We define the preconditions of a skill as a set of nodes and factors, thus considering a skill feasible *iff* the precondition factors are satisfied. For example, for state $s_0$ illustrated in Figure 1, the nodes of a factor graph are $\{L_0, H_0, R_0, B_0\}$ and the factors existing in this scene are $\{\texttt{Grasped}(L_0, H_0)\texttt{=True}\}$. Now, since this factor is a precondition of the skill $\texttt{Move}(L_0, H_0)$ that moves the hammer in hand to align with the box, it must be satisfied for the skill to be feasible. The effect of executing a skill creates a new factor graph $s'$ by changing the state of the nodes involved and, optionally, adding or removing their factors. This results in a *temporal factor* between the transitioned nodes of $s$ and $s'$ with the continuous action parameter of the skill $a_\pi$. The skill definitions can be extracted from standard PDDL symbolic skill operator with minor adaptations, following the duality of factor graphs and plan skeletons [1]. Eventually, we solve an optimization problem: satisfying the `Aligned`, `Grasped`, and the transition dynamics constraints by finding the correct `Move` parameters $a_{\pi_1}$. Each skill in a plan introduces additional nodes and factors to the factor graph, with added complexity for optimization.

Mathematically, we can use the distribution $p(s)$ as established in Equation 2 with all the spatial factors, and represent the temporal skill factor distribution of $k^{th}$-skill $\pi_k$ as the joint distribution: $p_{\pi_k}(s, a, s') \equiv p_{\pi_k}(S_{\pi_k}, a, S'_{\pi_k})$, $S_{\pi_k} \subseteq \mathcal{V}^{\pi_k}_{pre}$ which is executable *iff* the skill's pre-condition $s^{\pi_k}_{pre} \equiv \{\mathcal{V}^{\pi_k}_{pre}, \mathcal{F}^{\pi_k}_{pre}\}$ is satisfied by the current state i.e. $\mathcal{V}^{\pi_k}_{pre} \subseteq \mathcal{V}$ and $\mathcal{F}^{\pi_k}_{pre} \subseteq \mathcal{F}$. Once executed, it leads to the transitioned state $S'_{\pi_k}$. Based on the above formulation of a short-horizon transition distribution, we extend to construct a plan-level distribution as already established by GSC [6] and shown in Equation 1. We leverage the modularity of factored states by replacing states $s$ with a set of decision variables $S_{\pi_k}$ in the interest of skill $\pi_k$. This allows us to chain multiple skills in series and parallel. In such a scenario, the denominator term exists only for certain decision nodes *iff* they are common in two consecutive skills. We can indeed rewrite Equation 1 as:

$$p(\tau) \propto \frac{\prod_{\pi_k \in \Phi} p_{\pi_k}(v_k \in \mathcal{V}^{\pi_k}_{pre}, a_k, v'_k \in \mathcal{V}^{\pi_k}_{effect})}{\sqrt{\prod_{v_i \in \mathcal{V}_i} p_{\pi_{i-}}(v_i) p_{\pi_{i+}}(v_i)}} \tag{3}$$

if we consider that some set of intermediate nodes $\mathcal{V}_i$ are connected by two sequential skills $\pi_{i-}$ and $\pi_{i+}$.

**Representing coordination.** A key advantage of the factor graph representation is the ability to model multi-arm coordination tasks by connecting the temporal chains of each arm using spatial constraints. Such tasks often require skills to be simultaneously executed on each arm to operate

---

[1] i.e. a factor $f$ is included *iff* there is an edge between $f$ and some $v \in \mathcal{V}$ which also implies $v \in \mathcal{S}_f \subseteq \mathcal{V}$.

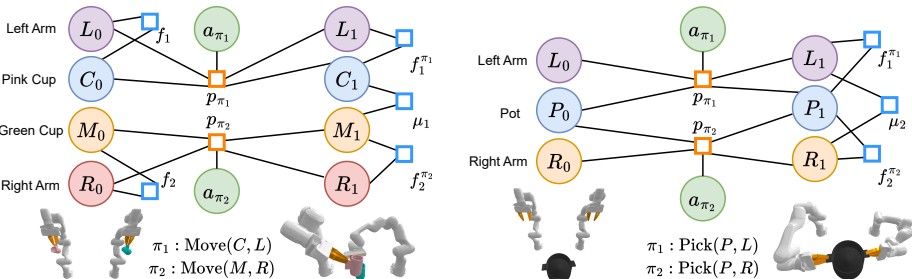

Figure 2: (Left) **Parallel independent chaining** The figure shows the execution of two skills ($\pi_1$ and $\pi_2$) in-parallel on two independent sets of nodes (L, C and R, M) to modify their existing factors (Grasped). The two independent executions can be connected via external factors $\mu_1$ (FixedTransform) introducing spatial dependencies between nodes C and M. (Right) **Parallel dependent chaining** The figure shows overlapping nodes of interest while parallel execution of two skills. The pot is to be picked by using both arms simultaneously. The effect of this is resulting factors (Grasped) between (L, P and R, P) and external factor $\mu_2$ (FixedTransform) between L and R. Overlapping nodes satisfy both skill's temporal effects.

on different or the same objects. We consider two cases for parallel skill execution, where multiple robots are operating on: (1) independent objects and (2) the same object, leading to independent and dependent temporal chains respectively. With our factorized state representation, we can independently control the execution of individual skills correlated with the nodes of interest and calculate the cumulative effect by applying the union of the effects of all the skills to the current factor graph. We consider a scenario shown in Figure 2 (Left). The left and right gripper arm $L_0$ are holding the pink $C_0$ and green $M_0$ cup ({Grasped($L_0, C_0$)=True} and {Grasped($R_0, M_0$)=True}) respectively. While both the grippers can independently execute the skill Move to modify separate factors ($f_1^{\pi_1}$ and $f_2^{\pi_2}$), one can add a constrained relationship factor ($\mu_1$) between the two mugs representing a set of transforms that satisfy the precondition of Pour. Such an ability to augment constraints flexibly allows zero-shot coordination planning for unseen tasks at test time even with parallel skill executions on the same object as shown in Figure 2 (Right).

### 4.2 Generative Factor Chaining

Now we have a formulation to construct a symbolic spatial-temporal factor graph plan for a task and chain them using spatial factor and temporal skill factors sequentially or in parallel. To make this plan geometrically feasible, we must find the optimal node variable values. We leverage the expressive generative model to capture the transition dynamics and exploit the compositionality of diffusion models. Given a symbolically feasible factor graph plan, our method, termed Generative Factor Chaining (GFC), can flexibly compose spatial-temporal factor distributions to sample optimal node variable values for the complete plan.

**Probabilistic model for trajectory plan as spatial-temporal factor graphs**. Now, we again consider the spatial graph for representing the state, where the probability of finding a state $s$ is the joint distribution of all the nodes in the factor graph. We will now integrate the spatial factors with the temporal factors considering the compensation term introduced in Equation 2 and Equation 3 along with the constraint factors across the chain $\mu \in \mathcal{M}$ as:

$$p(\tau) \propto \frac{\prod_{\pi_k \in \Phi} p_{\pi_k}(v_k \in \mathcal{V}_{pre}^{\pi_k}, a_k, v_{k+1} \in \mathcal{V}_{effect}^{\pi_k}) \prod_{k=0}^{K} \prod_{f \in \mathcal{F}_k} p_f(\mathcal{S}_f)}{\sqrt{\prod_{v_i \in \mathcal{V}_i} p_{\pi_{i-}}(v_i) p_{\pi_{i+}}(v_i)}} \Pi_{\mathcal{M}} f_\mu(S_\mu) \quad (4)$$

This completes the joint distribution of all the nodes in the spatial-temporal factor graph plan considering the temporal factors for all skills with their pre-condition and effect nodes, all spatial factors for all states in the plan, and all intermediate nodes in the temporal chain. It is worth noting that the augmented constraint factors $f_\mu$ work as a weighing function and can be more precisely represented by $f_\mu(S_\mu) \equiv f_\mu(y = 1|S_\mu)$ for some constraint-satisfaction index $y$.

We align towards diffusion model-based learned distributions to represent the probabilities in the formulated probabilistic graphical model. We transform the probabilities into their respective score functions $\epsilon(\mathbf{x}^{(t)}, t)$ for a particular reverse diffusion sampling step $t$ and train it using score matching

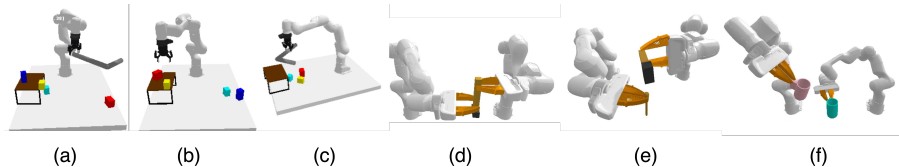

Figure 3: Evaluation tasks: **(a) Hook reach:** Hook is used to pull an object in the robot's workspace followed by other skills. **(b) Constrained packing:** Multiple objects must be placed on a rack without collisions. **(c) Rearrangement push:** Hook is used to push objects to a desired arrangement followed by other skills. **(d) Hammer place**: A hammer must be handed over to another manipulator and placed in a target box. **(e) Hammer nail**: A hammer must be handed over to another manipulator and a configuration must be achieved to strike a nail. **(f) Pour cup:** Cups must be brought in a configuration that allows successful pouring from one to another.

loss. Hence, for sampling a scene-graph for Equation 4, we have

$$\epsilon(\tau^{(t)}, t) = \sum_{\pi_k \in \Phi} \epsilon_{\pi_k}(v_k^{(t)} \in \mathcal{V}_{pre}^{\pi_k}, a_k^{(t)}, v_{k+1}^{(t)} \in \mathcal{V}_{effect}^{\pi_k}, t) + \sum_{k=0}^{K} \sum_{f \in \mathcal{F}_k} \epsilon_f(\mathcal{S}_f^{(t)}, t)$$
$$-\frac{1}{2} \sum_{v_i \in \mathcal{V}_i} \left[ \epsilon_{\pi_{i-}}(v_i^{(t)}, t) \epsilon_{\pi_{i+}}(v_i^{(t)}, t) \right] + \sum_{\mathcal{M}} \epsilon_{f_\mu}(S_\mu^{(t)}, t)$$

Such a representation leads to a cumulative score calculation of the joint distribution of all the nodes of interest to the factor using linear addition and subtraction. We can realize from Equation 4.2 that the final score function depends on the composition of all the factors in the spatial-temporal factor graph. While factors $f \in \mathcal{F}$ are mostly modeled implicitly by the temporal skills, the external factors can be any arbitrary spatial constraints that ensure the satisfaction of the pre-condition of the subsequent skills. Hence, with new additions to the set of external factors $\mu' \in \mathcal{M}'$, one can reuse the same temporal skills with added new spatial constraints. The proposed approach is modular as the individual skill factors and constraints can be flexibly connected to form new graphs. We have provided additional details in algorithm 1.

## 5 Experiment

In this section, we seek to validate the following hypotheses: (1) GFC relaxes strict temporal dependency to allow spatial-temporal reasoning, performing better or on par with prior works in single-arm long-horizon sequential manipulation tasks, (2) GFC can effectively solve unseen coordination tasks, and (3) GFC is adept in reasoning about long-horizon action dependency while being robust to increasing task horizons. We systematically evaluated our method on 9 long-horizon single-arm manipulation tasks from prior works and 4 complex multi-arm coordination tasks in simulation. We also demonstrate deploying GFC on a bimanual Franka Panda setup in the real world.

**Relevant baselines and metrics:** Our proposed method is based on factorized states and supports long-horizon planning for collaborative tasks directly at inference via probabilistic chaining. In this context, we consider prior methods based on probabilistic chaining with vectorized states (**GSC** [6]) and discriminative search-based approaches for solving long-horizon planning by skill chaining: with uniform priors (**Random CEM** or **RCEM**) or learned policy priors (**STAP** [5]). Since all prior works use sequential planning, we compare the performance of the proposed method on the sequential version of the parallel skeleton. Further information on data collection, training of skill diffusion models and real robot setup is provided in Supp. S4, Supp. S5 and Supp. S6 respectively.

**GFC relaxes strict linear dependency assumptions.** We first evaluate GFC on single-manipulator long-horizon tasks introduced by STAP [5] and also used by GSC [6]. These tasks consider manipulation by reasoning about the usage of a tool (a hook) to manipulate blocks out of or into the robot workspace (sample initial states shown in Figure 3(a-c)). While these tasks are originally designed to highlight linear sequential dependencies, there are steps with indirect dependencies or independence that only GFC can effectively model because of the factorized states. For example, in *Rearrangement Push*, the picking pose of the cube should not affect the tool use steps. As shown in Table 1, we observe that the performance of GFC is consistently on-par with the baseline for tasks with strict linear dependencies such as *Hook Reach* and on-par or better for tasks with more complex

Table 1: We show performance comparison of our method with relevant baselines on 9 single manipulator tasks and 3 two-manipulator tasks based on 100 trials for each of them. The task length shows the relative difficulty of solving them. We also conduct evaluation on 3 extended tasks to show robustness of GFC to task length ($|\mathcal{T}|$) and efficient reasoning about interstep dependencies.

| Evaluation Tasks | | | RCEM | DAF [4] | STAP [5] | GSC [6] | GFC | $|\mathcal{T}|$ |
|---|---|---|---|---|---|---|---|---|
| Single Manipulator | Hook Reach | T1 | 0.54 | 0.32 | **0.88** | 0.84 | 0.82 | 4 |
| | | T2 | 0.40 | 0.05 | 0.82 | **0.84** | 0.82 | 5 |
| | | T3 | 0.30 | 0.00 | 0.76 | 0.76 | **0.80** | 5 |
| | Rearrangement Push | T1 | 0.30 | 0.0 | 0.40 | **0.68** | **0.68** | 4 |
| | | T2 | 0.10 | 0.08 | 0.52 | 0.60 | **0.65** | 6 |
| | | T3 | 0.02 | 0.0 | 0.18 | 0.18 | **0.25** | 8 |
| | Constrained Packing | T1 | 0.45 | 0.45 | 0.65 | **0.75** | **0.75** | 6 |
| | | T2 | 0.45 | 0.70 | 0.68 | **1.0** | **1.0** | 6 |
| | | T3 | 0.10 | 0.0 | 0.20 | **1.0** | **1.0** | 8 |
| Bimanual Manipulation | Hammer Place | | 0.05 | - | 0.28 | 0.41 | **0.63** | 8 |
| | Pour Cup | | 0.10 | - | 0.18 | 0.15 | **0.41** | 4 |
| | Hammer Nail | | 0.02 | - | 0.15 | 0.15 | **0.34** | 11 |
| Longer Horizon Evaluation Tasks | | | | | | | | |
| Handback Hammer Nail | | | | | | | **0.24** | 16 |
| Handback Hammer Nail w/ auxilliary tasks | | | | | | | **0.25** | 18 |
| Handback Hammer Nail w/ extended auxilliary tasks | | | | | | | **0.21** | 20 |

dependency structures such as *Rearrangement Push*. This validates our hypothesis that GFC effectively models sequential dependencies, in addition to independence and skipped-step dependencies in long-horizon tasks.

**GFC can solve complex coordinated manipulation tasks.** Here, we aim to validate that GFC can effectively plan and solve different types of coordinated manipulation tasks. We present results on tasks with increased collaboration challenges. First, we consider tasks that require coordination but can be serialized into interleaved skill chains and solved by prior skill-chaining methods. *Hammer Place*, as shown in Figure S16, is for one arm to pick a hammer, hand it over to another arm for placement into a target box. *Hammer Nail* is an extension where, after hammer handover, first arm picks up a nail and both arms coordinate to move to positions such that the hammer's head is aligned with the nail for the subsequent striking step. The task is illustrated in Figure S16. As evident from Table 1, GFC significantly outperforms all baselines in both tasks. The gap is larger in the more challenging *Hammer Nail* task, which includes additional spatial and temporal constraints as shown in Supp. S7. This demonstrates that GFC can effectively model and resolve both spatial and temporal constraints in complex tasks.

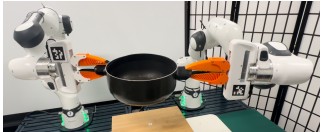

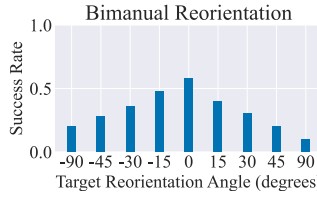

Figure 4: Evaluating GFC on bimanual reorientation where two arms simultaneously pick and reorient a pot.

**GFC can zero-shot generalize to new bimanual tasks by composing single-arm skill chains.** The *Pour Cup* (Figure S11) task is to Pick a cup with each arm, Move to position the two cups, and Pour the content of one into the other. GFC can directly reuse Pick and Move skill models and adapt the Strike skill model for the Pour step by adding a new spatial constraint. The constraint that "the cups can only be poured using the open top and not the closed bottom" can be directly added as a factor and optimized globally through guided diffusion process. A quantitative comparison is shown in Table 1. Finally, we consider the *Bimanual Reorientation* (Figure S12) task where two arms must simultaneously operate on the same object of interest (a pot), lift it up, and rotate it to a target reorientation angle (about z-axis) as illustrated in Figure 4 (Top) for a 45-deg angle. The tasks must be solved via parallel skill chaining with spatial constraints and hence none of the prior baselines can be used. The factor graph ( Figure 2 Right) includes a spatial fixed transform constraint between both the arms and hence the subsequent skills operate while satisfying the constraint. Figure 4 (Bottom) shows a detailed task success rate breakdown given different orientation goals.

**GFC can handle independence and inconsistent skill chains.** Here, we analyze how *independent* steps in a sequential manipulation chain affects the performance of each method. We consider *Hammer Place*, where the order of transporting the cube and handing over hammer is interchangeable. As illustrated in Figure 5, we consider a *consistent* plan skeleton where sequentially-dependent steps for the two main objectives, i.e., (1) opening lid then transporting cube and (2) picking, handing over,

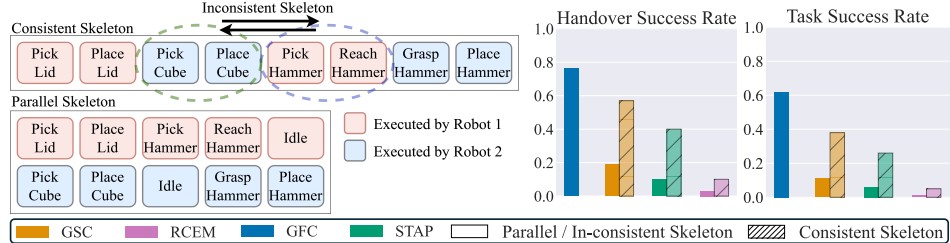

Figure 5: **Linear chaining has limitations.** Baseline methods with linear chain assumption suffers from performance drop when given inconsistent skill chains, where steps with sequential dependencies are swapped. GFC retains high success rate using the parallel skeleton.

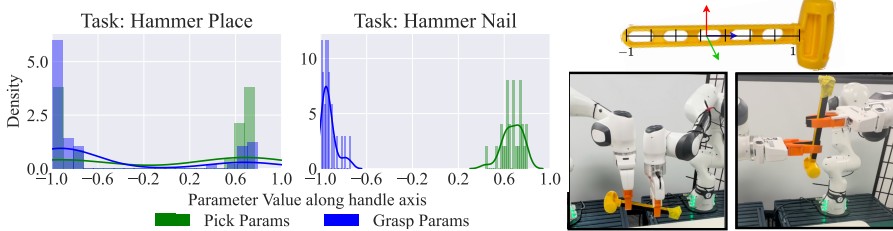

Figure 6: **Analysis of coordination.** We show that the planner is able to reason about the long-horizon action dependency of Pick and Grasp skills. (Left) While we see that *Hammer Place* can be solved by pick/grasp at head/tail and vice versa, to satisfy the precondition of Strike in *Hammer Nail*, the hammer must be grasped near tail so must be picked near head. (Right) We show orientation reasoning, where the hammer can either be grasped on the same side or the flip side.

and placing hammers, are completely sequentially. We also consider an *inconsistent* plan skeleton where the steps are interleaved. We show the handover success and overall task success in Figure 5 (Right). A successful handover requires choosing compatible parameters for Pick, Regrasp, and Move skills. While this increases the difficulty leading to lower scores in the handover success rate, the previous approaches failed to account for minor distraction and propagate the skipped-step dependencies as evident from the task success rate.

**GFC can reason about action dependency while being robust to increasing task horizons.** We observe in Figure 6 (left) that while *Hammer Place* task can be solved by picking or grasping on any end of the hammer handle, *Hammer Nail* requires more constrained parameter sampling. Further, in addition to the parameter selection along the handle axis, the method also samples suitable orientation (same or flip side) for grasping as shown by two examples in Figure 6 (right). We further give an example of the capability of our method in handling longer horizon inter-step dependencies in Figure S17 and simultaneously being robust with respect to the task length as shown in Table 1.

## 6 Limitations

First, our method does not generate high-level task plans. Solving the full TAMP problem with a unified generative model is an important future direction. Second, our method operates in a low-dimensional state space and hence requires a state estimator. We plan to extend GFC to work with high-dimensional observations. Finally, similar to prior works [4, 5, 6], our approach operates on parameterized skills.

## 7 Conclusion

We presented GFC, a learning-to-plan method for complex coordinated manipulation tasks. GFC can flexibly represent multi-arm manipulation with one or more objects with a spatial-temporal factor graph. During inference, GFC composes factor graphs where each factor is a diffusion model and samples long-horizon plans with reverse denoising. GFC is shown to solve sequential and coordinated tasks directly at inference and reason about long-horizon action dependency across multiple temporal chains. Our framework generalizes well to unseen multiple-manipulator tasks.

## 8    Acknowledgement

This project is funded by NSF Award 2409016 and the Georgia Tech AMPF Program.

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
