# OpenReview forum: "Generative Factor Chaining: Coordinated Manipulation with Diffusion-based Factor Graph"
_robot-learning.org/CoRL/2024/Conference — CoRL 2024_

### Official Review · Reviewer_tvkY · 2024-07-15
**Good work but lack many important details**

**Originality:** 3
**Technical Quality:** 4
**Clarity Of Presentation:** 3
**Potential Impact:** 3
**Recommendation:** 3
**Confidence:** 4

**Review:**

Pros
- Representing the planning problem as a spatial temporal factor graph is a good idea. The graph representation is more general than linear chains, which is adopted by a few previous approaches. The advantage is shown by the experiments also.
- The authors are clear about what are all the nodes in the graph. The factored state representation makes it easier for the model to learn and do inference.
- The idea of composing diffusion models to solve long-horizon problems is a good one. The current framework have some task level generalization theoretically.
- I appreciate the real world experiments.

Cons

While the idea is a good one, a few important details are missing out or not emphasized enough. The authors should also make it clearer what is the benefit to prior methods.
- The whole framework still requires a given graph structure, before the diffusion models are used to find the value assignment. This is a very important point but not emphasized enough. I hope the authors can make it clearer and bring it up early in the writing.
- Using factor graph as the representation is not new. It has been explored in the TAMP literature as pointed out by the authors. The major difference to me is differentiating the spatial nodes and the temporal nodes but I don’t see a clear benefit of that. Given a plan skeleton, the planning problem is a CSP and can readily be represented as one factor graph. If the authors want to highlight the representational advantage, it should be made clearer what is the benefit of having spatial and temporal factors.
- Not enough information about the diffusion models. Does one diffusion model correspond to a skill factor node or a spatial factor node, or either one? Also, the input are factored states and it’s assumed there is a state estimator giving all the required information. It should be made clearer too.
- The assumption on different part of the plan being independent is fine if the spatial constraints are not tight. If the task is to place blocks on a narrow shelf, then the placement of the first blocks will affect the placement of the second one, or the collision-free paths the arm can traverse.

**Quality Of The Limitations Section:**

3

**Questions For Rebuttal:**

- I’d appreciate more details on the diffusion models. e.g. Does one diffusion model correspond to a skill factor node or a spatial factor node, or either one? Are they trained jointly or separately? How well does it handle larger graphs? I imagine when the graph becomes larger, it’ll be harder for the individually trained diffusion models to all find a good sample.
- Generalization to new plan skeletons/graph structures?

**Robotics Focus:**

4

**Summary Of Paper:**

This paper proposes to use spatial temporal factor graph as a representation of robotics planning problems, called generative factor chaining (GFC). They use factored state representation and states are represented as variable nodes in the graph. Spatial constraint are represented using spatial factor node. GFC uses diffusion models as generative representation for each factor. For a planning problem, assuming the factor graph is given, GFC jointly optimize the diffusion models to obtain the final solution to the whole factor graph. The method is evaluated on both sim and real environments.

**Summary Of Recommendation:**

I think it's a good work but there are some important points that should be added or emphasized more in the writing. The difference to prior work with factor graph representation should be made clearer also. Update after the rebuttal: I think my concerns are addressed adequately. Recommendation updated.

---

### Official Review · Reviewer_FWri · 2024-07-20
**A factor graph approach for coordinated manipulation**

**Originality:** 3
**Technical Quality:** 2
**Clarity Of Presentation:** 2
**Potential Impact:** 3
**Recommendation:** 3
**Confidence:** 5

**Review:**

Overall, the approach is well-motivated and the concept of long-horizon task planning represented as a factor-graph has merit. The bimanual tasks demonstrated on real robots are challenging and relevant. However, the work is lacking sufficient details and clarity to understand the implementation of the factor graph which risks limiting the impact of the work.

**Strengths:**

* The experiments include challenging bimanual manipulation tasks which the method handles successfully.
* The authors compare against multiple baselines, which strengthens the claims.
* The use of the factor graph for TAMP is well-motivated and grounded in the literature.

**Points of feedback:**

The paper would benefit from a formal statement of the assumptions of the technique and the optimization problem being solved. This would help clarify how this method is more representative than existing methods, e.g. PDDL.  Notably, it is unclear what form the actions and states take (e.g. continuous or discrete), and how the inference problem stated in Eq. (1) is solved. The stated objective is to solve long-horizon planning problems, but the limitations section states that the proposed approach “does not generate high-level task plans.” A clearer formulation would help clarify the output of the method.

Some technical elements could use clarification. Specifically, how the connectivity of the factor graph itself is generated. It seems that the factor graph might encode both discrete (action type, grasped state) and continuous (object poses) variables, which would make this a non-trivial problem to solve (see for e.g. [1, 2]). Furthermore, the paper refers to constraints as probabilistic factors. It is notoriously challenging to formulate planning as inference with constraints, particularly in sampling-based planning (see [3]). How is this achieved in the proposed work?

The results would benefit from further analysis. Beyond success rate, it would be useful to know what causes failures. This is of particular interest for the tasks with low success rate, such as the longer-horizon evaluation tasks (bottom of Table 1), as this is a major claim of the work and the success rates are low, which contradicts hypothesis (3) (line 209). Additionally, the authors may consider including time to completion as a metric to demonstrate the parallel execution benefit.

The claims and methodology would benefit from more clarity in multiple areas. As currently written, it is difficult to follow the details of the implementation. Specifically:
* Multiple terms in the introduction are not defined, including “options” and “vectorized” states.
* The claim that previous works make a “linear” assumption is a bit unclear. Does this refer to the mathematical formulation of the motion planning problem or a sequential order of events?
* In the problem setup, Section 3, $\pi$, $a$, and $s$ should be related.
* In Section 4, it would be helpful to define vectorized and factorized states explicitly.
* In Table 1, T1, T2, and T3 are undefined and it is unclear how the task length is determined.
* The concepts of “consistent” and “inconsistent” skeletons is unclear. In Fig 5 for example, the inconsistent skeleton appears to be a subset of the consistent skeleton.

*Other points of feedback:*

* The figures of the factor graph are hard to parse due to the heavy use of notation. Figures 1 and 2 are very similar.
* Which skills are available? What spatial constraints exist? What assumptions are there about these skills and constraints?
* The equations relating to specific examples (Eq. (3) and the unnumbered equation after Eq. (4)) are hard to parse and could be moved to supplementary material.
* References 8, 43, and 46 are duplicates. (*Note: Added after official rebuttal period*)

**References**

[1] Yewon Lee, Philip Huang, Krishna Murthy Jatavallabhula, Andrew Z. Li, Fabian Damken, Eric Heiden, Kevin Smith, Derek Nowrouzezahrai, Fabio Ramos, Florian Shkurti. STAMP: Differentiable Task and Motion Planning via Stein Variational Gradient Descent. Conference on Robot Learning (CoRL), Learning Effective Abstractions for Planning Workshop, 2023.

[2] Alphonsus Adu-Bredu, Nikhil Devraj, Odest Chadwicke Jenkins. Optimal Constrained Task Planning as Mixed Integer Programming. International Conference on Intelligent Robots and Systems (IROS), 2022.

[3] Thomas Power and Dmitry Berenson. Constrained Stein Variational Trajectory Optimization. IEEE Transactions on Robotics, 2024.

**Quality Of The Limitations Section:**

2

**Questions For Rebuttal:**

The following are suggestions for the revised manuscript:

1. Clarify the problem formulation, assumptions, and methodology (especially how the factor graph is generated and how it is solved)
2. Clarify language and definitions of terms in the manuscript.
3. What accounts for the failures in the results?

See above for details and further feedback.

**Robotics Focus:**

4

**Summary Of Paper:**

This paper presents a factor graph-based technique for long-horizon task and motion planning. A representation which models states of all agents and objects as nodes and constraints and skills as factors is described. Experiments on simulated and real-world bimanual manipulation tasks demonstrate the success of the method.

**Summary Of Recommendation:**

The paper considers a challenging task and proposes a solution that performs better or on par to proposed baselines. There are multiple areas where the problem formulation, methodology, or results are unclear. Further, the results would benefit from more analysis to explain failures, particularly where they are inconsistent with claims.

---

### Official Review · Reviewer_dtQm · 2024-07-25
**Combines diffusion-based inference ideas for enforcing spatial- and temporal- constraints. Paper very well written with extensive experiments.**

**Originality:** 4
**Technical Quality:** 5
**Clarity Of Presentation:** 5
**Potential Impact:** 3
**Recommendation:** 4
**Confidence:** 5

**Review:**

## Strengths

- **Combines ideas from composable diffusion-based spatial constraint satisfaction and skill chaining.** The method combined approaches for composable diffusion based spatial constraint satisfaction from Diffusion-CCSP [25] and diffusion-based temporal skill chaining from GSC [6] in a unified framework. Using composable diffusion models in a factor graph framework is not new [25], however, as far as I’m aware, this is the first work that applies this framework to spatial *and* temporal constraint composition for long-horizon, multi-robot coordination.
- **Experiments are extensive.** The paper performs experimental evaluation on wide variety of tasks — 9 long-horizon single-arm manipulation tasks from prior works and 4 complex multi-arm coordination tasks. Experiments are perform both in simulation and the real world.
- **Experiments demonstrate strong generalization.** The experimental evaluation demonstrates strong generalization to unseen planning tasks with novel combinations of objects and constraints.
- **Effective bimanual manipulation:** Bimanual manipulation of a single, shared object is typically challenging due to the closed kinematic chain. It was satisfying to see in the supplementary video that the proposed method produces bimanual motion plans to manipulate a large cooking pot reasonably well on real robots, by automatically composing skills that were developed for single arms.
- **Paper is well written.** The paper is very well written, thoroughly organized and easy to follow.
    - **Introduction is well motivated.** The introduction motivates the gaps in prior work on skill chaining such as linear dependencies, and the necessity of spatio-temporal factor graphs multi-robot coordinated manipulation.
    - **Experiments section is very well organized.** The section begins with explicitly stating what hypothesis the paper is seeking to validate, and explicitly states the conclusions and key findings from the experiments.
    - **Useful illustrations and running examples.** The running example of the hammering task is very helpful for the user to understand the proposed method. The factor graph illustrations in Fig. 1 and Fig. 2 also help explain the proposed concepts effectively. It was very helpful for Fig. 2 to show examples of factor chaining when (1) robots independently operate separate objects, and (2) when they operate a common object.
    - Appendix Sec. S4 provides a helpful and illustrative explanation of the proposed method.
- **Paper described limitations of the work.** Paper clearly describes the limitations of the work (not integrated with TAMP, uses object state etc.) and the supplementary video shows failure cases of bimanual manipulation of the large pot.

## Weaknesses

- **Factor graph inference algorithm can be simplified by treating spatial and temporal factors similarly.**
    - The proposed method treats factors representing spatial constraints (at the same timestep) *differently* from skill factors representing temporal constraints. The former uses the technique from Diffusion-CCSP where score functions of spatial factors are simply added (based on Eq. 2). The latter uses the forward-backward analysis of GSC [6] which introduces additional terms in the denominator of the probability density (Eq. 3). It seems to me that this separate treatment is awkward and unnecessary. The method can be simplified by just adding the score functions of both spatial and temporal factors following Diffusion-CCSP without additional terms in the denominator.
    - The paper attempts to justify “denominator compensation” in the Appendix Sec. S3 lines 64-75. However the provided justification seems problematic. Specifically, once the expression for $p(\tau | s_0)$ and $p(\tau | s_1)$ are written down (between lines 72 and 73), Eq. (S3) combines them by assuming  $p(\tau | s_0, s_2) \propto \sqrt{p(\tau| s_0) \ p(\tau | s_2)}$. However, it is unclear how this can be theoretically justified. Instead, the expression should be:

        $$
         \begin{align*}
          p(\tau | s_0, s_2) &\propto p(s_0, \tau, s_2)
          \propto p(\tau | s_0)\ p(s_2 | \tau)
          \propto \frac{p(\tau | s_0)\ p(\tau | s_2)}{p(\tau)}
          \end{align*}
        $$

         My point is that “denominator compensation” seems inaccurate and ad-hoc to begin with, and is probably unnecessary for the method.

    - My recommendation is to conduct experiments where denominator compensation is removed, and evaluate whether it is necessary for empirical performance. If not, doing away with denominator compensation will not only simplify the proposed method, but it will also treat spatial and temporal factors in a more consistent and unified way.

- **Framework does not support constraints during skill execution.**
    - The framework supports constraint satisfaction at discrete timesteps, corresponding to the beginning and end of skill executions. This allows applying collision constraints to ensure that the robots are not colliding with other objects only at mode switches (corresponding to states $ \{ s_0, s_1, s_2, \dots \} $ ). But the method cannot guarantee that robots won’t collide *during* the execution of the skills. This is because individual skills are learned independently and modularly (in the absence of other objects and robots) and are executed independently.
    - Similarly, when performing bimanual manipulation of the pot, relative pose constraints between the pot and two end-effectors can only be enforced at the beginning and end of the trajectory. But the closed kinematic chain requires that the relative pose constraints be satisfied *throughout* trajectory execution. How is/can this be achieved in the proposed framework?

## Minor weaknesses

- It would be useful to explicitly cite which prior works the 9 single-arm manipulation tasks (e.g. *Hook Reach*, *Rearrangement Push* etc.) are derived from.
- Typo in Line 90: “multi-model distributions” should be “multi-modal distributions”.

**Quality Of The Limitations Section:**

3

**Questions For Rebuttal:**

Please discuss and respond to the issues raised in the “weaknesses” section of the review.

**Robotics Focus:**

4

**Summary Of Paper:**

The paper presents an approach for planning long-horizon trajectories for multi-step, multi-arm manipulation by composing multiple diffusion models in a factor graph framework. The planning problem is represented as factor graph [8]. The nodes of the factor graph are states of robots and objects that are either known or are treated as decision variables be optimized. Factors represent spatial and temporal constraints between nodes (such as pre- and post- conditions of skills, relative pose constraints etc.). Each diffusion model is trained to learn individual factor types. Then, the score function of the joint probability distribution is composed of score functions of individual diffusion factors. This enables modular, composable inference to generate long-horizon plans that show strong generalization to unseen tasks and novel combinations of objects and constraints. An extensive experimental study on simulated and real robots demonstrates that the proposed method solves sequential and coordinated multi-robot tasks such as handover maneuvers, hammering nails etc. with superior performance and generalization compared to existing baselines.

**Summary Of Recommendation:**

Overall, the paper combines approaches for composable diffusion based spatial and temporal constraint satisfaction in a unified framework, and applies it to challenging long-horizon, multi-robot coordination. The paper is very well written. The experimentation is extensive and demonstrates good generalization to novel combinations of objects and constraints. Notwithstanding some concerns about simplifying the inference algorithm, this is a good paper and I recommend a strong accept.

---

### Author Rebuttal · Authors · 2024-08-09

We would like to thank the Area Chair and all the reviewers for their thoughtful comments and valuable feedback. We have carefully addressed and replied to each of the individual comments in the respective reviews. Additionally, we have attached the revised manuscript and supplementary, with all changes marked in blue.

---

### Decision · Program_Chairs · 2024-09-04

**Decision:**

Accept

**Comment:**

Strengths:
1. Integrates ideas from composable diffusion-based spatial constraint satisfaction and skill chaining in a unified framework, extending the application to long-horizon, multi-robot coordination.
2. Demonstrates comprehensive experimental evaluation on a variety of tasks, both in simulation and real-world settings, showcasing the method's generalization to unseen tasks.
3. Successfully handles challenging tasks like bi-manual manipulation of shared objects, demonstrating practical utility.
4. The paper is thoroughly organized, with a well-motivated introduction, clear illustrations, and a detailed explanation of the methodology and experiments.
5. Clearly addresses the limitations of the work, including the lack of integration with TAMP and constraints on object states.

Weaknesses:
1. The separate treatment of spatial and temporal factors seems unnecessary and could be simplified, as the current method introduces complexity without clear justification.
2. The framework supports constraints only at discrete timesteps, not throughout skill execution, potentially leading to collisions during motion.
3. Requires a predefined graph structure before applying the diffusion models, which is not sufficiently emphasized or justified.
4. Lack of clarity on whether diffusion models correspond to skill or spatial factor nodes, and how they handle larger graphs.
5. Questions remain about the method's scalability to larger graphs and tighter spatial constraints.